# Potential Application of Protamine for Antimicrobial Biomaterials in Bone Tissue Engineering

**DOI:** 10.3390/ijms21124368

**Published:** 2020-06-19

**Authors:** Michiyo Honda, Morio Matsumoto, Mamoru Aizawa

**Affiliations:** 1Department of Applied Chemistry, School of Science and Technology, Meiji University, Kanagawa 214-8571, Japan; mamorua@meiji.ac.jp; 2Department of Orthopaedic Surgery, School of Medicine, Keio University, Tokyo 160-8582, Japan; morio@a5.keio.jp

**Keywords:** protamine, antimicrobial peptide, hydroxyapatite, biofilm formation, implant-related infection

## Abstract

Bacterial infection of biomaterials is a serious problem in the field of medical devices. It is urgently necessary to develop new biomaterials with bactericidal activity. Antimicrobial peptides and proteins (AMPs), alternative antibacterial agents, are expected to overcome the bacterial resistance. The aim of this study was to develop a new intelligent material in bone tissue engineering based on protamine-loaded hydroxyapatite (protamine/HAp) that uses AMPs rather than antibiotics. It was found that the adsorption of protamine to HAp followed the Langmuir adsorption model and was due to electrostatic and/or hydrophobic interactions. In vitro bacterial adhesion and growth on protamine/HAp was inhibited in a protamine dose-dependent manner. Adherent bacteria exhibited an aberrant morphology for high dosages of protamine/HAp, resulting in the formation of large aggregates and disintegration of the membrane. The released protamine from protamine/HAp also prevented the growth of planktonic bacteria in vitro. However, a high dosage of protamine from powders at loading concentrations over 1000 μg·mL^−1^ induced a cytotoxic effect in vitro, although those exhibited no apparent cytotoxicity in vivo. These data revealed that protamine/HAp (less than 1000 μg·mL^−1^) had both antimicrobial activity and biocompatibility and can be applied for bone substitutes in orthopedic fields.

## 1. Introduction

Implants have been used to improve loss of function, replace lost tissue, or optimize appearance. Various medical devices with different raw materials and shapes have been developed. Hydroxyapatite (HAp) is the principal inorganic constituent of hard tissues (bone and teeth) and has been widely applied in tissue engineering as an orthopedic and dental material due to its biocompatibility and osteoconductivity. Furthermore, HAp can form very tight bonds with living bone, resulting in the promotion of new bone formation. In many researches, HAp has been employed as bioactive coatings onto titanium [1] and polymer substrates [2] for enhanced skeletal fixation. Calcium phosphates, mainly as HAp, have been used in the field of bone tissue engineering, though the low bioresorbability of sintered HAp resulted in the cause of implant-related infection. On the other hand, HAp is also widely used as a column material in affinity chromatography for the separation of various proteins because of its ability to absorb proteins, amino acids and other substances [3,4,5]. This means that bacteria can also prefer to adhere to the HAp surface and form biofilms. To avoid the incidence of implant-related infections, several approaches have been employed [6,7,8]. In particular, surface modifications with antibacterial coatings, immobilized molecules, or light-activated molecules are widely explored novel strategies [9]. While antibiotics are still essential for the success of surgical procedures, the increase in bacterial resistance to conventional antibiotics has necessitated the development of alternative therapies.

Causative pathogens are reported to be *Staphylococcus aureus*, coagulase-negative *staphylococci*, *Enterococcus* spp., and *Escherichia coli* in surgical site infections (SSIs), which were defined as infections by organisms occurring up to 30 days after surgery [10,11]. There is a need to find alternatives with a broad spectrum of activity against several species of bacteria.

Antimicrobial peptides and proteins (AMPs) have attracted much attention as a promising alternative to antibiotics. AMPs are the host defense peptides and are key elements of innate immunity [12]. Targeting of the bacterial membrane and the concomitant loss of membrane integrity is a well-accepted mechanism of action of AMPs [13,14]. In addition, several reports demonstrated that AMPs can disturb a series of cellular processes and metabolic functions [15]. AMP’s multiple modes of action would be superior to antibiotics, which act only on one specific target. These modes of action enable reducing the bacterial drug resistance [16].

Protamine, which is an arginine-rich protein extracted from fish milt, is a promising potential AMP. Protamine is a cationic AMP that inhibits or kills many microorganisms, including bacteria and fungi. The electrostatic interaction between (positively charged) protamine and (negatively charged) bacterial membrane is suggested to be the driving force for the expression of the protamine bactericidal activity. However, the precise mechanism of action of protamine at the molecular level remains unclear. Additionally, protamine, biomedical application, has been focused in the field of tissue engineering and regenerative medicine. Basic protamine molecules complexed with acidic molecules such as heparin, flagmin, and DNA form complexes via ionic interaction. Nano/microparticles were generated by electrostatic interaction and can function as immobilizers or attractants of various cytokines. Those nano/microparticles have been applied as biomaterials such as cell carriers, protein carriers, and growth factor carriers [17,18].

In this study, to develop a new bone substitute with antimicrobial activity, protamine was applied to biomaterials. We evaluated the bactericidal properties of protamine-loaded HAp (protamine/HAp) and demonstrated its suitability for use in bone tissue engineering.

## 2. Results

### 2.1. Preparation and Characterization of Protamine/HAp Powders

To prepare the protamine/HAp powders, HAp powders were mixed with various concentrations of a protamine solution for 48 h and were adsorbed by the batch method. The protamine adsorption to HAp powders increased with increasing the concentration of protamine in a dose-dependent manner (Figure 1) and reached saturation at over 1000 μg·mL^−1^ protamine. The adsorption of protamine to HAp followed the Langmuir adsorption isotherm.

An examination of the XRD patterns of each protamine/HAp powder showed no obvious changes due to protamine adsorption with only HAp single phase peaks observed (data not shown).

Next, to investigate the properties of the protamine/HAp powders, the zeta potential was measured. The zeta potential of the protamine/HAp powders also increased with increasing the concentration of protamine (Figure 2a) and reached a plateau at over 1000 μg·mL^−1^ protamine. As shown in Figure 2b, the two sets of results (adsorption levels of protamine to HAp powder and their zeta potential) were positively correlated. Furthermore, the surface of the protamine/HAp sample was examined by X-ray photoelectron spectroscopy (XPS). The XPS survey spectra (data not shown) indicated major peaks at N, C, 0, Ca, and P that originated from HAp and protamine. The adsorption of protamine to HAp can be evaluated from the N 1s band spectrum. These data indicate that protamine mainly adsorbed HAp via electrostatic interactions.

### 2.2. Evaluation of Antimicrobial Activity of Protamine/HAp Discs

To assess the antimicrobial activity of protamine/HAp powders, protamine/HAp discs were fabricated by uniaxial compression of the powders. *Escherichia coli* (*E. coli*) and *Staphylococcus aureus* (*S. aureus*) were seeded on different types of protamine/HAp discs and cultured in Luria-Broth (LB) medium for 24 h. At 24 h after plating, the ability of protamine/HAp discs to affect the viability of *E. coli* and *S. aureus* was tested using the BacLight LIVE/DEAD staining kit (Figure 3). The fluorescence microscope images of *E. coli* and *S. aureus* on the HAp disc showed that the surface of the disc was covered by adhered living bacteria (green), and these bacteria started the formation of a biofilm (Figure 3a,g). However, the number of living bacteria (green) decreased and that of dead bacteria (red) increased with the increasing protamine concentration (Figure 3b–f,h–l). Especially, the number of living *S. aureus* decreased drastically on each protamine/HAp disc, resulting that *S. aureus* were more sensitive than *E. coli* to protamine (Figure 3g–l). For the living *E. coli* on the protamine/HAp discs, it was observed that bacteria induced the aggregation and formation of clusters. In addition, an aberrant morphology of bacteria (elongated *E. coli*) on the protamine/HAp discs was observed (Figure 3c,d). These data demonstrated that the *E. coli* on the protamine/HAp discs showed inhibited cell division or DNA replication. Furthermore, to investigate the morphology of the bacteria on the protamine/HAp discs in more detail, *E. coli* on the discs were observed by SEM (Figure 4). The bacteria on the HAp or protamine/HAp (125 μg·mL^−1^) appeared as normal rod-like shapes (Figure 4a,b). By contrast, the morphology of *E. coli* on the protamine/HAp discs (doses of 250 and 500 μg·mL^−1^) showed elongated shapes (Figure 4c,d). High dosages of protamine/HAp discs (over 1000 μg·mL^−1^) caused the formation of large aggregates (Figure 4e,f). These phenomena were also observed in the supernatant of protamine/HAp (Appendix A). These results revealed that bacteria can easily adhere to the surface of HAp and form biofilms (Figure 4g).

On the other hand, positively charged protamine/HAp discs electrically attracted the negatively charged *E. coli* and reduced the bacterial motility (Figure 4h). As for *S. aureus*, no apparent changes were seen in their morphology, though adhered cells dramatically decreased (Appendix A).

Next, to clarify the mechanism of antimicrobial properties of protamine/HAp on E. coli, the concentration of the protamine released into the LB medium was measured. It was found that the concentration of protamine in the LB that was released from the discs increased with increasing amounts of adsorbed protamine (Figure 5, solid circles: -●-). However, the protamine released from all discs was approximately 7–10% of the total adsorbed amount of protamine. The amount of the bacteria in the LB medium was affected by the protamine concentration (open circles: -○-). A low concentration of protamine (~250 μg·mL^−1^) induced cell aggregation and led to an increase in the number of bacteria. By contrast, a high dosage of protamine (over 500 μg·mL^−1^) induced cell death. These data indicate that the bactericidal ability of protamine/HAp discs was mainly due to the release of protamine from HAp. Interestingly, the release of protamine from discs only occurred in the ion-rich media such as the LB medium and simulated body fluid. Analysis of the ion concentration in the LB medium showed that the concentration of sodium ions in LB decreased due to the release of protamine (data not shown). These results suggest that the adsorbed protamine was released through ion exchange.

### 2.3. Antimicrobial Susceptibility of Protamine

As described above, the antimicrobial activity of protamine/HAp was mainly due to the release of protamine from HAp. To evaluate the antimicrobial susceptibility of protamine against bacteria, microtiter broth dilution method [19] were carried out. As shown in Figure 6, protamine was found to prevent the growth of *E. coli* dose-dependently. Inhibition of microbial growth by 50% was induced by protamine in concentrations ranging from 100 to 200 μg·mL^−1^, and bacterial growth was completely prevented by the protamine concentration of 400 μg·mL^−1^ (Figure 6a, solid circles: -●-). In S. aueres, a treatment of protamine over 50 μg·mL^−1^ inhibited bacterial growth (Figure 6b, solid circles: -●-). These results indicate that protamine could prevent both Gram-negative and Gram-positive bacteria, and *S. aureus* was more sensitive to protamine than *E. coli*. The effect of protamine on the membrane surface charge of bacteria was assessed by zeta potential analyses. In Figure 6 (open circles: -○-), the bacteria in the absence of protamine showed a zeta potential of −20.26 ± 0.84 mV for *E. coli* and −10.55 ± 1.37 mV for *S. aureus*, respectively. Treatments with protamine induced the neutralization of the negative charge and stabilized the zeta potential at approximately 3 mV for *E. coli* and 0 mV for *S. aureus*, indicating the entrapment of the protamine beyond electrostatic equivalence.

### 2.4. Evaluation of Biocompatibility of Protamine/HAp Discs

Generally, antibacterial ability conflicts with biocompatibility. Therefore, to evaluate the biocompatibility of protamine/HAp, in vitro cytotoxicity tests were carried out according to ISO 10993-5:2009 “Tests for Cytotoxicity—In vitro Methods” [20]. Human osteoblastic cells (MG-63) were cultured in extracts from various kinds of powders (0.1 mg·mL^−1^) for 24 and 72 h. As shown in Figure 7, the results of the MTT assay at 24 and 72 h showed that no significant changes of cell viability were seen in concentrations ranging from 125 to 500 μg·mL^−1^. However, the viability of the cell culture exposed to a high dosage of protamine/HAp (over 1000 μg·mL^−1^) induced cell death. These data indicated that released protamine from protamine/HAp powders (over 1000 μg·mL^−1^) was toxic for osteoblasts. On the other hand, extracts at dilutions of 50% and 25% (50 and 25 μg·mL^−1^) did not negatively affect the viability at 24 and 72 h (data not shown).

Furthermore, to evaluate the biocompatibility of the protamine/HAp (0 vs. 1000 μg·mL^−1^) in vivo, the specimens (diameter: 4 mm, height: 8 mm) were implanted into the tibiae of Japanese rabbits for eight weeks (Figure 8). Histological observation by Villanueva bone staining showed that the newly formed bones were directly in contact with both HAp and protamine/HAp discs without fibrous tissues (shown by arrowheads in Figure 8b,d). It was concluded that HAp and protamine/HAp discs were compatible with the rabbit tibia and that no degradation of the implant material occurred at the intervals of up to eight weeks after the implantation. No significant difference was found between the biological behavior of the discs prepared using HAp only and the discs prepared using the protamine/HAp powders. Taken together, these results demonstrate that the protamine/HAp powders (by the concentration at 500 μg·mL^−1^) were both osteoconductive and biocompatible in vitro and in vivo.

## 3. Discussion

As previous antibiofilm therapeutics have been unsatisfactory, current research has focused on preventive strategies and, particularly, on implant surface modifications. Several strategies, namely the antiadhesive, contact-active, and biocide release strategies, can be pursued to obtain antimicrobial coatings by chemical modifications [9]. Among these, we focused on AMPs [13,21] and fabricated the novel protamine-loaded HAp biomaterial that inhibits biofilm formation. HAp is most frequently used as a bone-substitute material in orthopedic and dental fields because of its similarity to bone in chemical composition and its direct bonding to bone.

The levels of the adsorption of protamine to HAp powders increased with increasing protamine concentrations in a dose-dependent manner and essentially followed the Langmuir adsorption isotherm. The zeta potential of the protamine/HAp powders also increased with increasing protamine concentrations. This meant that adsorption occurred because of the electrostatic interaction between the amine groups of arginine and the hydroxyl and/or phosphate groups of HAp. In this study, isotropic HAp was used; however, the negatively charged c-planes of HAp can be expected to adsorb large amounts of protamine [22]. Furthermore, HAp with a fine structure and a large specific surface area will be effective for increasing the loading amount of protamine.

Next, to investigate the interaction between protamine/HAp and bacteria, viability tests using protamine/HAp discs were performed (Figure 3). *Escherichia coli* adhered and formed biofilms on HAp, although the number of living bacteria decreased. Interestingly, the formation of large aggregates was observed for high dosages of protamine/HAp discs (over 1000 μg·mL^−1^). For the intermediate levels of protamine (500–1000 μg·mL^−1^), we observed elongated *E. coli* that could not divide (Figure 4). This phenomenon was also observed in planktonic bacteria (Appendix A). Bacterial elongation, called filamentation, was also obtained by treatment with indolicidin [23] that can permeabilize the bacterial membrane and bind to DNA, resulting in the inhibition of DNA synthesis. Protamine can also bind to DNA and form DNA/protamine complex by electrostatic forces, which was applied to gene delivery [17]. The interaction of protamine and DNA also prevents DNA synthesis [24,25].

As for the release profile of protamine, the amount of the released protamine increased with the increase in the adsorbed protamine concentration (Figure 5). Protamine adsorbs HAp through electrostatic and/or hydrophobic interactions. HAp has an ion-exchange ability for the exchange of cations with the calcium ion sites and the exchange of anions with the phosphate groups or hydroxide ion sites. In fact, changes in the salt concentration in the LB medium affected the desorption of protamine. These data imply that the release of protamine from HAp can be controlled by adjusting the salt concentration in the surrounding environment. Furthermore, a previous investigation reported that proteins can adsorb in a multilayer fashion on the HAp surface. The increase in the distance due to the multilayers weakened the electrostatic interaction between the protein molecules and HAp [26]. Thus, we observed the easy release from protamine/HAp (over 1000 μg·mL^−1^) that was higher than the saturated adsorption level. This phenomenon was also observed in our previous study [27]. These results indicate that bacteria were attacked by both immobilized and released protamine. The use of protamine/HAp(500) would be a good candidate for an antibacterial bone substitute, because released protamine from protamine/HAp(500) could damage the bacterial membrane and lead to cell death. HAp enables to release protamine by ion-exchange for the long term. In addition, it is expected that using bioresorbable materials such as tricalcium phosphate (TCP) make it possible to release protamine simultaneously with the resorption of materials by osteoclasts.

As previously reported [13,14,28], AMPs, including protamine, kill bacteria by acting on bacterial membranes or cell walls. Generally, AMP is amphiphilic and positively charged and has both hydrophobic and hydrophilic parts [12,16]. Positively charged cationic peptides can interact with negatively charged cell membranes via electrostatic interactions and adsorb the membrane. After binding to the membrane, AMPs can form pores and disintegrate the membrane bilayer structure, leading to bacterial death. Several models of membrane permeation by peptides were provided, such as barrel-stave pore, toroidal pore, carpet model, and detergent model [21]. In this study, to investigate the interaction between protamine and the bacteria, the zeta potential of the bacterial cell membrane was measured. Our data showed that protamine exhibited antibacterial activity in a concentration-dependent manner against both Gram-negative and Gram-positive bacteria (Figure 6 and Appendix A). Additionally, the surface potential of the bacteria was positively shifted by the treatment with protamine. These results agree with the results of previous studies [27,29]. The positively charged AMPs interact with the negatively charged cell membranes through electrostatic interactions and undergo membrane adsorption and conformational changes. LIVE/DEAD staining also showed that the bacteria adhered on the protamine/HAp disc (over 500 μg·mL^−1^) were dead due to the disruption of the membrane by the attached protamine. Additionally, we observed the differences of susceptibility to protamine between *E. coli* and *S. aureus*, which might be caused by the differences of the bacterial membrane structures. In the present study, S. aureus were more susceptible to protamine, and a low concentration of protamine (100 μg·mL^−1^) could disrupt the membrane, leading to cell death. In contrast to *S. aureus*, protamine would be able to past the cell wall without causing damage and bind to DNA in *E. coli* [30]. The inhibition of the DNA synthesis of *E. coli* changed their morphology without achieving cell division. However, the precise mechanism of action of protamine remains unknown, and further studies are needed to clarify other mechanisms.

With regards to the biocompatibility of AMPs, it was previously found that AMPs can selectively bind to the outer surface of the negatively charged bacterial cell membranes without interacting with the outer surface of the neutral eukaryotic cell membranes [13]. In our study, a high dosage of released protamine exhibited a cytotoxic effect. In contrast, the in vivo assessment also showed that protamine/HAp can directly bond to bone (Figure 8). There are some differences, such as protamine concentration, cell numbers, and cell types, between in vitro and in vivo situations. The discrepancy between the in vitro and in vivo biocompatibility of protamine resulted from blood circulation. Protamine has already been clinically applied as a heparin antagonist and, therefore, is known to be safe [31]. In medical use, protamine sulfate is normally administered at the dose of 10–15 mg·mL^−1^ [32]. Therefore, the concentration of protamine in this study would be biocompatible. However, since pathogenic microbes become peptide-resistant after long-term use, the side effects of long-term use should be investigated. Taken together, our results show that protamine-loaded medical devices can inhibit biofilm formation and reduce the incidence of implant-associated infections.

## 4. Materials and Methods

### 4.1. Preparation and Characterization of Protamine-HAp Powders and Discs

Various concentrations of protamine sulfate solutions (125–2000 μg·mL^−1^) were diluted by phosphate-buffered saline (PBS). HAp powder (1.5 g; HAp-100; Taihei Chemical Inc., Osaka, Japan) were added into each protamine solution (45 mL) and incubated for 48 h at room temperature. The samples were centrifuged for 15 min at 8.000 rpm, and the supernatant was assayed for the protein using the Bio-Rad (Hercules, CA, USA) protein assay following the manufacturer’s instructions. Powders were washed in PBS 5 times and freeze-dried. To fabricate the discs (diameter: 15 mm, thickness: 1–2 mm), the resulting powders (0.30 g) were uniaxially compressed at 50 MPa.

Measurements of zeta potentials of protamine/HAp powders were carried out using a laser-Doppler velocimeter (ELS-6000; Otsuka Electronics, Osaka, Japan) in a 10 mM NaCl solution at room temperature. The zeta potential was calculated from the measured electrophoretic mobility. The complete experiment was repeated for a total of three separate assays. The morphological observation of the sample powders was performed at an accelerating voltage of 15 kV by scanning electron microscopy (SEM; JSM-6390LA; JEOL, Tokyo, Japan).

### 4.2. Antimicrobial Evaluation

To evaluate the antibacterial activity, Escherichia coli (*E. coli*, ATCC 27325) and Staphylococcus aureus (*S. aureus*, ATCC 6538) were used in this study. Bacteria were grown in a culture broth (nutrient broth from LB; Wako, Osaka, Japan) and culture agar (nutrient agar from LB; Wako, Osaka, Japan) at 37 °C in an incubator.

#### 4.2.1. Antimicrobial Susceptibility Tests

Using the microtiter broth dilution method [19], the antimicrobial susceptibility tests of protamine against *E. coli* and *S. aureus* were performed. In brief, protamine (Maruhanichiro Inc., Tokyo, Japan) was solubilized in PBS to a final concentration of 100 mg∙mL^−1^ and sterilized by filter with a 0.22-μm pore size. Protamine solutions then were diluted to yield final concentrations of 0.5, 1.0, 2.0, and 4.0 mg mL^−1^. Dilutions were dispensed (0.02 mL/well) in the wells of a polypropylene microtiter plate already containing 0.16 mL/well of the prepared 1 × 10^6^ cfu mL^−1^ bacteria inoculum and 0.02 mL/well of Alamar blue (Invitrogen, Carlsbad, CA, USA). Final protamine concentrations of 50, 100, 200, and 400 μg∙mL^−1^ were tested. Each concentration was tested in triplicate. Samples were incubated for 18 h at 37 °C without shaking. The bacterial growth in suspension was monitored at 570 nm and 600 nm. The relative cell viability (%) was calculated by a ratio of optical density (OD)_sample_/OD_control_ × 100 for each of the values. The experiment was repeated for a total of three separate assays.

#### 4.2.2. Measurement of the Surface Charge of Bacteria

The zeta potential analyses were carried out using an ELSZ-1000 (Otsuka Electronics, Osaka, Japan) at room temperature. Protamine solutions were prepared at the concentrations ranging from 50 to 400 μg·mL^−1^ using 10 mM 4-(2-hydroxyethyl)-1-piperazineethanesulfonic acid (HEPES) buffer solution (pH 7.4), containing 150 mM NaCl. Protamine stock dilution (0.1 mL) was mixed with 0.9 mL of the bacterial suspension. The bacterial suspensions were dispensed into microtubes and incubated for 15 min at room temperature. The zeta potential was calculated from the electrophoretic mobility. The complete experiment was repeated for a total of three separate assays.

#### 4.2.3. LIVE/DEAD Staining

Bacteria cultured on various concentrations of protamine/HAp discs were examined for cell viability using a BacLight LIVE/DEAD staining kit (Invitrogen, Carlsbad, CA, USA). Equal volumes of SYTO^®^ 9 and propidium iodide (PI) were mixed in microtube and then the dye mixture solution was added to PBS. Following resuspension of each sample in 1 mL of the staining solution, samples (bacteria on the discs or in the supernatants) were incubated for 15 min in the dark at room temperature. After washing with PBS, samples then were imaged using fluorescence microscopy (IX71; Olympus, Tokyo, Japan). Each concentration was tested in triplicate.

#### 4.2.4. Measurement of the Cell Concentration by Optical Density

Optical density (OD), measured in a spectrophotometer (Gene Quant 100; GE Healthcare, Piscataway, NJ, USA), can be used as a measure of the concentration of bacteria in a suspension. Bacterial concentrations were determined by measuring the OD at 600 nm. The OD values of the inoculum were standardized to OD = 1.0 at 600 nm (approximately 8 × 10^8^ CFU·mL^−1^).

### 4.3. Release Profiles of Protamine from Protamine/HAp Discs

The protamine/HAp discs with different protamine contents were placed on 24-well plates (BD, Franklin Lake, NJ, USA), which were filled with bacterial suspension at 37 °C. After 24 h culture, concentrations of protamine released from discs were then measured using the Bio-Rad protein assay.

### 4.4. Biocompatibility Evaluation

#### 4.4.1. Evaluation of Biocompatibility In Vitro

Human osteoblasts, MG-63 cells, were obtained from ATCC^®^ CRL-1427^TM^ (Manassas, VA, USA). MG-63 cells were cultured in Eagle’s minimum essential medium (MEM; Sigma, St. Louis, MO, USA) supplemented with 10% heat-inactivated fetal bovine serum (Sigma, St. Louis, MO, USA) and 0.1% antibiotics (penicillin-streptomycin solution, Sigma) in a humidified atmosphere containing 5% CO_2_ at 37 °C. According to ISO 10993-5:2009 “Tests for Cytotoxicity—In vitro Methods” [20], extracts were obtained in a separate culture medium (0.1 g·mL^−1^ of the culture medium for 24 h at 37 °C). For cytotoxicity test, cells were seeded at 5 × 10^3^ cells·well^–1^ on 96-well plates and cultured for 24 h at 37 °C/5% CO_2_. Each extract (0.1 mL) was then applied to cultured cells. After 24 h and 72h of incubation, culture medium was replaced with a fresh medium. Subsequently, 5 mg·mL^−1^ of MTT solution (10 μL) was added, and cells were cultured for 4 h at 37 °C/5% CO_2_. The MTT solution was then removed, and formed formazan crystals were solubilized by DMSO (100 μL) per well. After that, the absorbances at 570 nm (measurement wavelength) and 650 nm (reference wavelength) of each well were quantified by a microplate reader (Multiskan FC; Thermo Fisher Scientific, Waltham, MA, USA). The relative cell viability (%) was calculated by a ratio of OD_sample_/OD_control (HAp at 24 h)_ × 100 for each of the values.

#### 4.4.2. Evaluation of Biocompatibility In Vivo

To assess the biocompatibility and osteoconductivity of protamine/HAp discs in vivo, 16-week-old male rabbits (average weight: 3 kg) were used in this study. The specimens (diameter: 4 mm, height: 8 mm) were sterilized by ethylene oxide gas. The tibia of a rabbit was exposed, and cylindrical defects (4.2 mm in diameter) were drilled in the epiphysis of the tibia. The HAp or protamine/HAp discs were then implanted into the defect for 8 weeks. After implantation, the rabbit was sacrificed using sodium pentobarbital, and the tibia was removed. For undecalcified histological analysis, harvested implants were fixed 70% ethanol, dehydrated in an alcohol series, defatted, embedded in methylmethacrylate, and cut into 7 μm sections using a microtome. After staining with Villanueva bone stain, these specimens were observed under light microscopy (IX71; Olympus, Tokyo, Japan). All experiments were approved by the Keio University Institutional Animal Care and Use Committee (09067-(10)).

## 5. Conclusions

We developed a novel biomaterial with bactericidal coatings using protamine in this study. Protamine, a cationic peptide, is adsorbed to HAp through electrostatic and/or hydrophobic attraction. Both the protamine immobilized on HAP and the protamine released from HAp inhibited the adhesion and early growth at the material’s surface. These protamines are likely to disintegrate the bacterial membrane and bind to DNA, resulting in the inhibition of DNA synthesis. The findings reported in the present study suggest that protamine can be effectively used for antimicrobial materials. Protamine-loaded calcium phosphates such as HAp and tricalcium phosphate are believed to have great potential for use as antimicrobial biomaterials in bone tissue engineering.

## Figures and Tables

**Figure 1 ijms-21-04368-f001:**
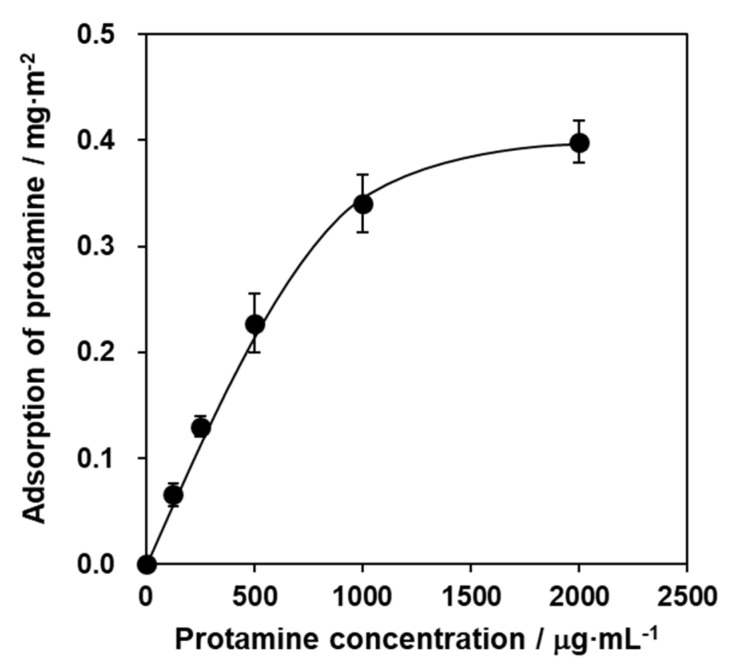
Adsorption levels of protamine to hydroxyapatite (HAp) powders. HAp powders were mixed with various concentrations of protamine solutions for 48 h. Adsorption of protamine was determined by residue of protamine in supernatants using the Bio-Rad protein assay. Each value represents the mean of triplicate tests. Error bars indicate the standard error of the mean.

**Figure 2 ijms-21-04368-f002:**
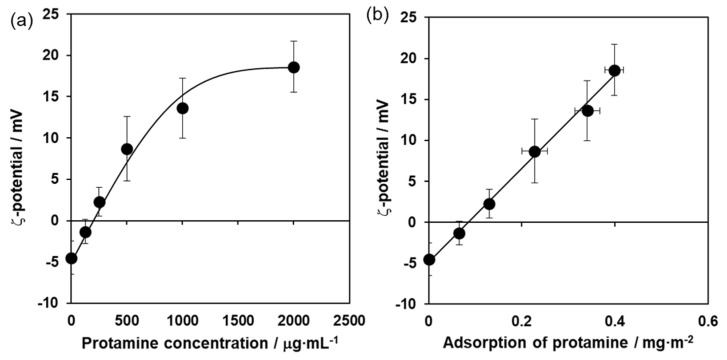
Zeta potential properties of protamine-adsorbed HAp powders. Zeta potential of protamine/HAp powders were measured (**a**). Adsorption of protamine and its zeta potential were related to a positive correlation (**b**). Each value represents the mean of triplicate tests. Error bars indicate the standard error of the mean.

**Figure 3 ijms-21-04368-f003:**
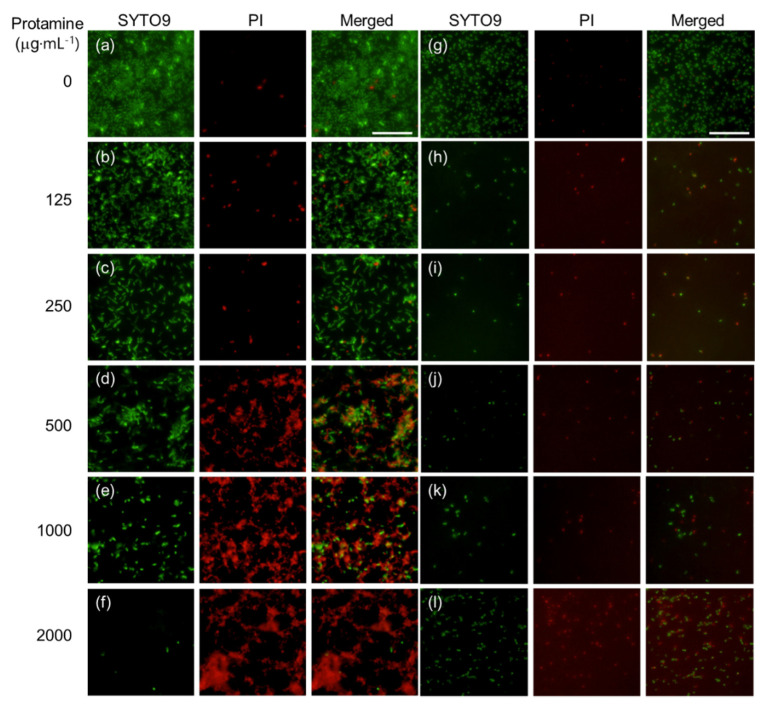
LIVE/DEAD staining of bacteria on the surface of protamine/HAp discs. *Escherichia coli* (**a**–**f**) and *Staphylococcus aureus* (**g**–**l**) were cultured on various concentrations of protamine/HAp discs: (**a**,**g**) 0 (HAp), (**b**,**h**) 125, (**c**,**i**) 250, (**d**,**j**) 500, (**e**,**k**) 1000, and (**f**,**l**) 2000 μg∙mL^−1^ for 24 h. Cells were stained using the LIVE/DEAD Bacterial Viability kit, as described in Materials and Methods, and observed by fluorescence microscopy (live/dead = green/red, respectively). Bars indicate 20 μm.

**Figure 4 ijms-21-04368-f004:**
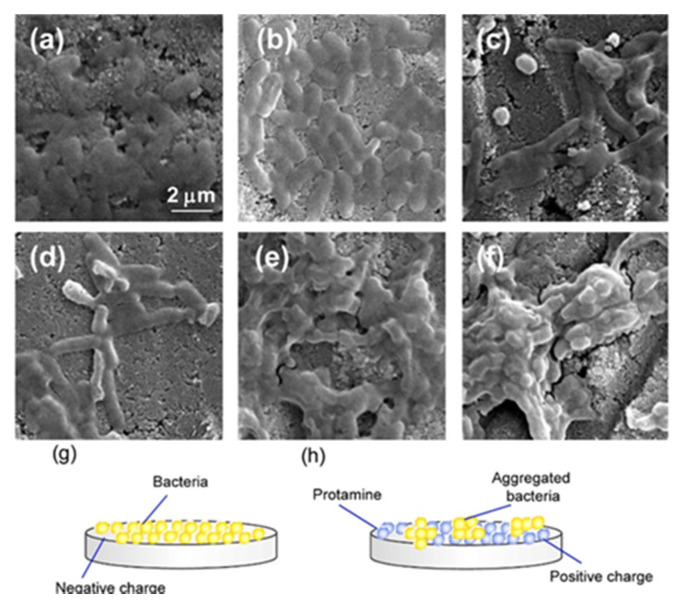
SEM images of *E. coli* cells on the surface of protamine/HAp discs. *E. coli* cells cultured on various concentrations of protamine/HAp discs: (**a**) 0 (HAp), (**b**) 125, (**c**) 250, (**d**) 500, (**e**) 1000, and (**f**) 2000 μg∙mL^−1^ for 24 h were observed by SEM. Bar indicates 2 μm. Schematic diagram of the adhesion of bacteria to HAp (**g**) or protamine/HAp disc (**h**).

**Figure 5 ijms-21-04368-f005:**
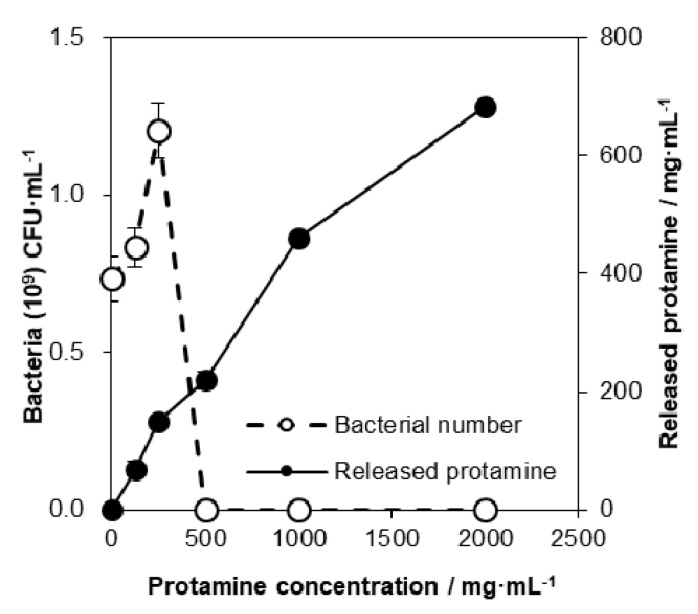
Relationship between released protamine and bacterial number. *E. coli* cells were cultured on various concentrations of protamine/HAp discs for 24 h. The released protamine into Luria-Broth (LB) medium was measured by the Bio-Rad protein assay. The number of bacteria in supernatants after 24 h culture was determined by measuring the optical density (OD) at 600 nm.

**Figure 6 ijms-21-04368-f006:**
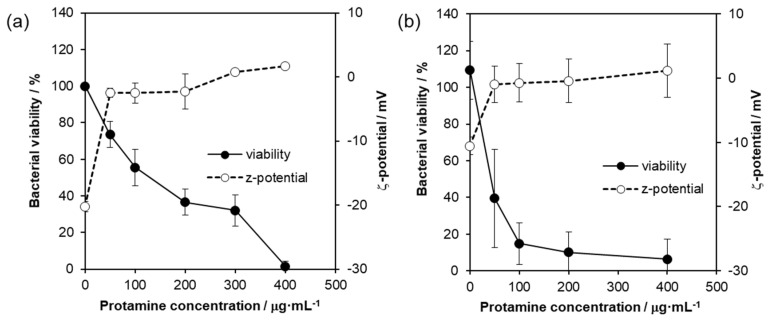
The bacterial viability and zeta potential of bacteria by treatment of protamine. *E. coli* (**a**) and *S. aureus* (**b**) were treated with protamine for 18 h. Various concentrations of protamine were tested. Solid circles (-●-) indicate the % of viable bacterial cells by treatments of protamine, while the zeta potential is represented by the open circles (-○-). Each value represents the mean of triplicate tests. Error bars indicate the standard error of the mean.

**Figure 7 ijms-21-04368-f007:**
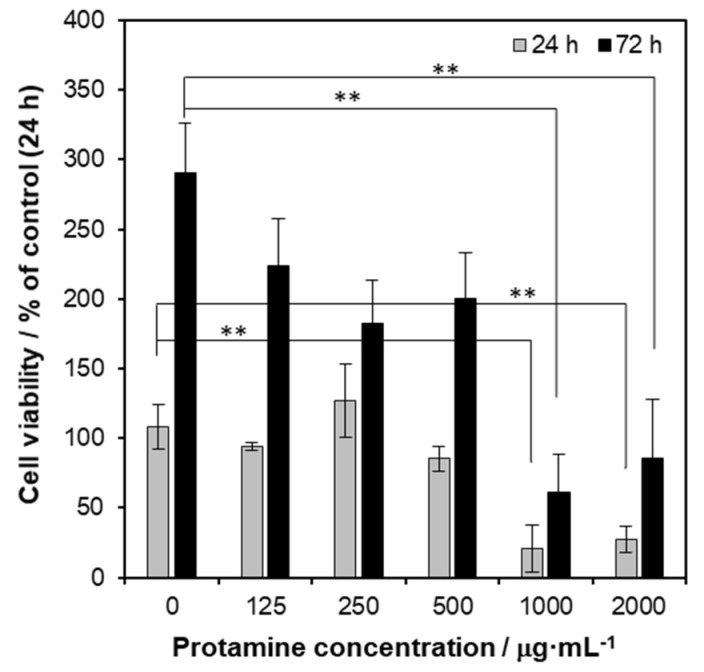
In vitro biocompatibility of protamine/HAp powder extracts to osteoblastic cells. MG-63 human osteoblastic cells were cultured in various concentrations of protamine/HAp powder extracts for 24 and 72 h. Cell viability was analyzed using ISO 10993-5:2009 “Tests for Cytotoxicity—In vitro Methods”.

**Figure 8 ijms-21-04368-f008:**
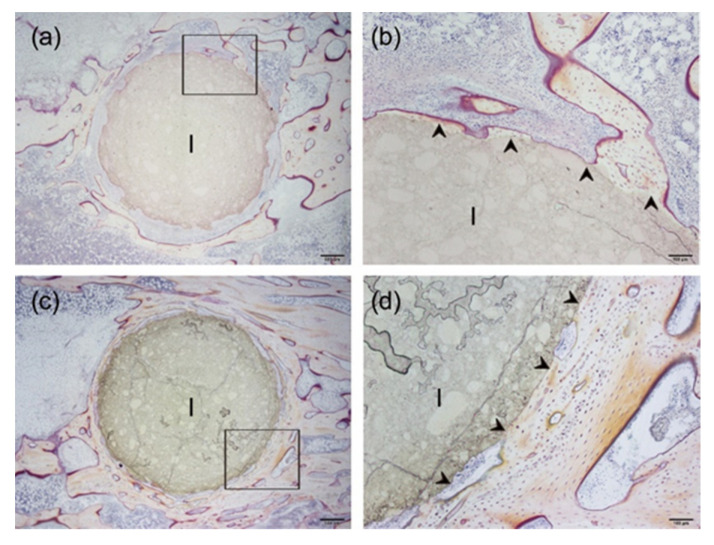
In vivo biocompatibility of protamine/HAp discs in a rabbit tibia. HAp (**a**,**c**) or protamine/HAp (**b**,**d**) discs were implanted in rabbit tibiae for 8 weeks. Villanueva bone staining of each specimen (I) showed the newly formed bones (arrowheads) were directly in contact with discs. (**b**) and (**d**) are highly magnified images of (**a**) and (**c**), respectively. Bars in (**a**) and (**c**) indicate 500 μm, and (**b**) and (**d**) indicate 200 μm, respectively.

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
