# Peer review of "Potential Application of Protamine for Antimicrobial Biomaterials in Bone Tissue Engineering"

_ijms, 2020, doi:10.3390/ijms21124368_

Round 1
Reviewer 1 Report
The MS presented by Honda et al. sounds interesting because the authors intend to test a new use of the natural polypeptide - protamine, as component of modified biomaterial. Until now, this product has been used in medicine for many purposes, whose scope has been expanding for several years by more and more new indications, including dentistry and regenerative medicine.
Research on the potential use of protamine in dentistry was previously carried out (published in 2019) by the authors of the present MS. It should be mentioned that these two MS are very similar in terms of aim, main idea to prepare dicalcium phosphate (then) or hydroxyapatite (now) loaded with protamine, as well as methodology of antimicrobial activity testing of these complexes.
If so, the authors should better justify the choice of the same peptide for research into the idea of its other use, and should emphasize the originality of their ideas more strongly.
It is well known that this fish-derived peptide possess some antimicrobial activity, but since such a product has been tested before, the authors should better emphasize the new approach to the problem and underline the obtained original information that goes beyond the earlier available reports.
Publication by Nemeno et al. (2014 ) entitled "Applications and Implications of heparine and protamine in tissue engineering and regenerative medicine" and by Ishihara et al. (2015) entitled "Biomedical application of low molecular weight heparin/protamine nano/micro particles as cell- and growth factor-carriers and coatings matrix" (unfortunately not cited), should be discussed.
The current MS version is not devoid of some limitations. The authors should consider the following general and specific comments.
Title - The MS title, however is attractive, but in the light of the research presented, it is too vague. Question - which justifies the use of the term "intelligent medical device".
Abstract
L9-14 - it is too long general (vague) introduction to study presented. The abstract should be concise and informative: what and why it was tested, what methods have been used, what results were obtained and what conclusions they derive. It is suggested to re-write this.
L17-18 - ...Bacterial adhesion ......... should be added in vitro,
L21-22 - the same comment as above,
L23-25 - this statement goes far beyond the presented research.
Section 1. Introduction - does not sufficiently state the need for this study.
- The authors should define clearly and precisely the direction chosen for the study.
- The authors should better describe the use of hydroxyapatite in the medical area (except dentistry) and should absolutely complete present epidemiological and microbiological data of infections in such area. Only then E. coli included in the study could be justified.
- Some of the information is unnecessarily placed in the Discussion section, from where it should be moved to Introduction and detailed.
Through this, MS can gain some scientific value and this would allow to better emphasize the importance of the shown in vivo effects.
L56-59 - this statement goes far beyond the presented research
Section 4. M&M
Subsection 4.1. and 4.2. should be combined and shortened since the same is described in previous author's publication (Fuiki M. ....... Honda M. Materials, 2019)
Subsection 4.3. - certain limitations on the validity of tests which cause the inability to repeat these tests in another laboratory are:
- the use of only one bacterial strain E. coli (probably clinical isolate) and the lack of strain characteristics; no explanation for the lack of inclusion of the reference strain (e.g. from the ATCC collection);
- no justification for the advisability of such a choice in the context of the "orthopedic" direction of research;
- no information concerning S. aureus strain used
- no explanation is given for not using a standard protocol in line with the Clinical and Laboratory Standards Institute (CLSI) (2012) guideline to determine protamine MIC / MBC concentrations.
- L 294-305 - description is unclear - please use the standard units for concentration of tested products micrograms/miligrams per mL), density of bacterial inoculum (CFU/mL) etc. The same units should be used in M&M, Results (Tables, Figures) and Discussion sections,
- L323 - the calibration curve for OD600 and density dependency of the E.coli suspension should be presented (I will return to this issue in the commentary on the resulting part [L153-160].
Section 2. Results
subsection 2.1
L63-65 - The Authors wrote that: ..."Protamine was found to be inactive against the gram-positive bacterium Staphylococcus aureus (S. aureus) at the concentrations below 25 mg·dm3 (data not shown). Therefore, S. aureus was excluded from the rest of the study".
- The statement requires a thorough explanation. This result (below 25 mg·dm3 ) is much more corporal than that obtained for E. coli, since protamine was active against E. coli in concentrations 200 mg/dm3) and 400-500, causing 50% and approx. 90-100% inhibition, respectively.
- What, finally, were the MIC / MBC values of protamine determined by the standard broth microdilution assay, and how does it relate to concentrations up to 1000-2000 mg / L (?), used further in the study. It is necessary for the authors to present direct results (OD600 values) from the study described in subsection 4.3.1 (M&M) for these two bacterial species, not just intermediate results (%).
subsection 2.3.
The Authors should consider shortening the descriptive part of the results obtained in this series of experiments. Graphic documentation and a synthetic description of the most important observations would be sufficient. In addition, results comments in this subsection could be deleted because they are not their place.
subsection 2.4.
- The authors conclude about the biocompatibility of protamine for osteoblast cells deposited on modified discs, based on their fluorescent staining and morphology assessment and wrote that (L167-169): ..."no obvious changes in the cell morphology were observed on each disc, demonstrating that protamine/HAp was biocompatible in vitro. The released protamine in the culture medium did not affect cell viability".... It seems, however, that it would be necessary to perform a classical in vitro cytotoxicity test, determining IC50 and calculating the biocompatibility index (BI), i.e. the relationship between cytotoxicity and microbicidal activity of protamine (according to e.g. Müller and Kramer, 2008). This is the standard approach in research into new antimicrobial "solutions" for potential clinical use.
L169-175 - The authors should explain here first, that not such disks as those tested in vitro, were used in in vivo studies (15 mm x 1-2 mm versus 4 mm x 8 mm). Hence, the results obtained in in vitro tests for the former, do not necessarily translate directly into the use of the latter. In addition, the M&M procedure does not describe the preparation of 4 mm x 8mm disks implanted in rabbits. This aspect of research, with clinical context, needs to be documented.
Section 3. Discussion
This part is definitely too long. It contains many of the same information as in the Introduction section and repeats of the description of results. However, in many places the interpretation of the important observations made is missing. It should be rewrite.
L188-193 - This is an unnecessary piece of text (repetition of basic and obvious knowledge).
L199-200 - The authors wrote that: ..." These data showed that protamine exhibited antibacterial activity in a concentration-dependent manner against both gram-positive and gram-negative bacteria (Fig. 1).", which is not true. In the M&M section this species was not given as the study object, and in the Results section it was stated that S. aureus was excluded from the study - in my opinion, for unjustified reasons = three places of text - three different information.
L204-205 - According to the authors:: .... "These data revealed that protamine is an effective candidate alternative agent for antibiotics". This statement is not authorized by the tests performed or documented by data from other publications.
L230-237 - What the authors define as a biofilm and what mechanisms of elongation of E. coli are mentioned - it is suggested to confront these observations more deeply with the research of other authors.
L251-256 - This description of AMPs' mechanisms of action is highly incomplete - it should be supplemented.
Section 5. Conclusions
This section is rather summary of the results. Final conclusion (last sentence is vague and obvious).
L366-367 - This statement is not supported by the research conducted by the authors.
List of references
The correctness of bibliographic data requires careful checking - there are many errors.
Reviewer 2 Report
It was a pleasure to read your manuscript. The results are promising for potential application of AMPs as therapeutics. This is very important that protamine saves its activity after binding to hydroxyapatite and there is a hope that could be applied safely.
I have only two issues which need to be explained/ clarified:
- Please describe more precisely why did you choose E.coli vs. Staph. aureus for the study. You are working with bone related material and staphylococci are main cause of bone/biofilm related infections. Therefore I don't undersrand why higher activity against Staph was the reason not to continue witg this strain.
- Please add some quantification data for the biocompatibility in vitro assay. Basing on the visualisations it is nit so obvious that there is no impact of protamine on the cells. Some numbers/plots with statistics would highly improve this part of study.
Reviewer 3 Report
Antimicrobial peptides are a good approach to increase implant safety and reduce implant-related infections. The authors present promising results using their protamine-loaded hydroxyapatites. However, the manuscript is not ready for publication.
Title
The term “tissue engineering” has no place in the title of this manuscript. There is no application mentioned in tissue engineering.
Results
Line 64: The authors mention that they also studied S. aureus but results are not shown. I could not understand why they excluded this data from the manuscript, especially when a gram-positive bacterium is missing.
Figure 4: I suggest to add the protamine concentrations on the left side for each group in the figure. It would be much easier for the reader. In this case, the authors performed LIVE/DEAD staining of the biofilms on the disks. However, they did not quantify the differences between the groups. This could be done by using a confocal laser scanning microscope and subsequent quantification of live and dead proportions using appropriate image quantification software. In this case, also the volume of the biofilm could be measured. It is really important to quantify data to prove the antibacterial activity of the novel implant material!
Figure 6: It is not really clear how the experiment was performed. Was E. coli just added on top of the disks? Isn’t it the same as the one in Figure 4? If the authors want to show the toxicity of released protamine they should use the extracts of disks and perform a classic antibacterial test like CFU counting.
Biocompatibility (Figure 8): Actin filament and nucleus staining are not enough to prove cytocompatibility. Techniques allowing quantification should be used. For example, the metabolic activity of cells (MTT, cell-titer-blue), cytotoxicity (LDH release), etc. These pictures only show the adherence of cells and their morphology. No conclusion about proliferation could be drawn, as mentioned by the authors in line 270. In addition, some of the pictures have really high background in the nucleus channel (blue).
Discussion
Line 192: “Biofilms formed by bacteria are…” --> which bacteria? This comment also applies to the introduction. Which bacteria are associated with implant infections and why was only E. coli studied?
Line 203: “…against both gram-positive and gram-negative bacteria (Fig. 1).” --> Only results for E. coli are shown, which were not adequately quantified, thus, the statement is wrong.
Line 259: “In this study, it was also found that protamine binds to the bacterial membrane through electrostatic interactions…” --> where is the data to prove this?
Overall, the approach for a novel material is good. Nevertheless, there are serious concerns like the lack of quantifications and of appropriate methods to prove antibacterial activity and concomitant biocompatibility. As for the biocompatibility, the authors should also consider human cells, since they behave totally different. Especially, immortalized cell lines like MC3T3 are not sensitive enough compared to human primary cells. Another issue is the use of only one bacterial species. The authors do not explain the complexity of implant infections and related species. This study tested the novel material superficial and not deep enough to prove the efficacy. I encourage the authors to conduct additional experiments, revise the manuscript deeply, and to present a completed scientific story. It needs just a little bit more effort.
Good luck!
Round 2
Reviewer 1 Report
I accept the amendments. They cover most (but not all) of my comments and concerns.
I have no further serious comments.
However, in the interest of obtaining the highest level of publication, I suggest making a few more minor corrections and additions:
1). L39-48 and L51-55 - the content is chaotic, repeated and vague.
2). L61-62 - this rather important information should be somewhat expanded, since as many as two publications related to it are cited.
3). L127 - word Next should be not at italic
4). L128 -144 - this description need to be discussed more thoroughly in the Discussion section (L222-233 and L 258-261) since it is well known that the risk of bacteria acquiring resistance to a low concentration of a given antimicrobial is a serious therapeutic problem.
5). L240 - why uppercase letters are used.
6). L278 - please correct "E. coil" to E. coli.
7). L278- 279 - please explain strain collection name (?) of E. coli K12W3110 and S. aureus IAM1011.
8). L284-287 - is it true that the initial protamine solution was prepared at a concentration of 100 microg/ml, which was later diluted to obtain final solutions at a concentration of "0.5, 1.0, 2.0, and 4.0 mg/ml" - how is this possible?
9). L 287-288 - ... "0.02 ml/well) in the wells of a polypropylene microtiter plate already containing 0.16 cm3/well of the prepared 1 × 106 cfu∙ml-1 bacteria inoculum and 0.02 ml/well" - these units should be harmonized.
10). L 353 - the abbreviation HAp TCP should be provided in full since it is not on the abbreviation list (L361).
11). the reference list format is still patchy and needs improvement.
Reviewer 3 Report
The manuscript profited from the additional results and the scientific story is more prominent. Nevertheless, I would improve the text, especially the discussion, to increase the impact of the manuscript.
Title
I would remove the term intelligent and replace it by antibacterial to provide a more accurate title. The proposed material has antibacterial properties. It is not intelligent. Intelligent biomaterials would be responsive to a situation like an infection.
General
The English have to be improved. For example, take care of proper use of “on the other hand”. Someone should mention first “on the one hand” and then to give a contrast “on the other hand”. Also, proper use of prepositions like in line 145 “Antimicrobial Susceptibility of Protamine” à should be “to”.
Bacterial species should be written in italics.
Discussion
The discussion has to be revised to improve the manuscript. The results should be discussed in relation to work from other scientists and also in relation to each other.
Some missing points:
Discussion of different response of S. aureus and E. coli, why is one more susceptible than the other and what does it mean for the implant success, does this have an influence? What does it mean that E. coli is elongated and S. aureus not?
In the revised version, the authors added results regarding the in vitro biocompatibility. There is a cytotoxic effect at a certain concentration, where would be the therapeutic window? This means, which concentration would cover appropriate antibacterial activity and biocompatibility? How is this in relation to the in vivo data? This a little bit discussed in the abstract, but not in the main discussion. What is expected to happen if the implants are infected in vivo? There, could be some interesting observations in the literature, only one publication is cited regarding the cytocompatibility of protamine.
The manuscript improved by adding more data and providing deeper investigation, however it would tremendously benefit from a thorough discussion connecting all data and discussing it more deeply with the current literature.
Best wishes!
Round 3
Reviewer 3 Report
Dear authors,
your work is worth publishing, you did a lot of experimental work! I have an advice for you. Please, consider for the future to improve your writing and to discuss deeper your results by putting them in context with previous work on this topic. It might be that you need to read more scientific articles.
All the best